# Effects of Physical Activity on Cognitive Functioning: The Role of Cognitive Reserve and Active Aging

**DOI:** 10.3390/brainsci13111581

**Published:** 2023-11-11

**Authors:** Giulia D’Aurizio, Fabiana Festucci, Ilaria Di Pompeo, Daniela Tempesta, Giuseppe Curcio

**Affiliations:** Department of Biotechnological and Applied Clinical Sciences, University of L’Aquila, 67100 L’Aquila, Italy; giulia.daurizio@univaq.it (G.D.); fabiana.festucci@graduate.univaq.it (F.F.); ilaria.dipompeo@graduate.univaq.it (I.D.P.); daniela.tempesta@univaq.it (D.T.)

**Keywords:** active aging, physical activity, cognitive functions, elderly, psychophysical well-being, cognitive reserve

## Abstract

Background: The increase in average life expectancy necessitates the identification of possible mechanisms capable of promoting “active aging” to ensure adequate levels of global functioning. Numerous studies show that regular physical activity promotes, even in the elderly, a state of functional psychophysical well-being capable of slowing down age-related cognitive decline. This study aimed to clarify whether, and how, the intensity of physical activity can modulate cognitive and executive skills by influencing specific psychological variables. Methods: Our sample consisted of 151 senior subjects divided into hikers (HIK), gentle gymnastics (GYM), and sedentary (SED), who practice intense, moderate, and reduced physical activity, respectively. A battery of psychological questionnaires was administrated to evaluate attentional skills, decision-making, the ability to implement targeted behaviors, perceived self-efficacy, and psychophysical well-being. We included: the Mini-Mental State Examination, Cognitive Reserve Index Questionnaire, General Self-Efficacy Scale, Letter Cancellation Test, Everyday Competence Questionnaire, and Geriatric Depression Scale (GDS). Results: Comparisons between the scores reported by the three groups showed that the HIK group differs from the others with respect to most of the measurements, presenting better mood and cognitive performance, and a specific psychological profile. On the contrary, the GYM group appeared to have a greater affinity with the SED group than with the HIK group, both cognitively and psychologically. Conclusions: Types of physical activity, as well as the intensity and frequency with which they are practiced, are factors that promote an active aging process, protecting the psychophysical well-being and overall cognitive functioning of the elderly.

## 1. Introduction

The progressive increase in life expectancy is producing significant social, economic, and health consequences, changing our way of looking at the elderly: a positive perspective is taking hold in which aging is seen in terms of success, well-being, and resources. Aging can be defined as a process or set of processes that take place in a living organism and which, over time, decrease its chances of survival [1]. Old age is therefore a phenomenon characterized by physical and mental changes, not due to illness, which involves a reduced ability to adapt to stress and maintain homeostatic balance [2]. Typically, physiological aging is characterized by a progressive cognitive decline, but the evolution of this alteration and the consequent individual differences can be mediated by numerous factors [3]. This individual variability seems to depend on variables that can speed up or slow down age-related cognitive impairment [4]. In light of this, cognitive reserve [CR] represents a protective factor in the neurodegenerative process, describing a complex strategy with which the brain tries to resist damage secondary to physiological or pathological aging through the intervention of brain plasticity processes and rearrangement of cognitive functions [5,6,7]. Changes in physical appearance are the most evident, resulting in a progressive modification of the body image that can lead to negative psychological implications if not adequately processed and reintegrated into the self-image. With the biological aging of the central and autonomic nervous systems, the weakening of sensory abilities and the raising of perceptual thresholds are inevitable. Since the sensory organs are fundamental in our relationships with external, physical, and social environments, elderly subjects can feel as though they are losing their functionality and autonomy, and this increases the subjective perception of being old. No less important are alterations in the sense of balance and the kinesthetic sense, which contribute to an increased risk of falls, with a consequent loss of motor autonomy. In the brain, reductions in volume and weight, gray matter, and dendritic arborization can be observed [8]. At the neurochemical level, changes in the production and reuptake of glutamate and dopamine have been reported. Decreases in blood flow and oxygen and glucose consumption are also observed, resulting in various dysfunctions and slowdowns in some cognitive processes [9,10]. Memory is considered a central aspect of cognitive aging; however, if some of its components tend to worsen, others maintain high levels of efficiency. Procedural memory appears to be relatively preserved, while only slight impairments in short-term memory, semantic memory, and autobiographical memory are recorded; the most important impairments are instead highlighted in the domains of working memory, episodic memory, and prospective memory [11,12,13]. The difficulties encountered in aging during the performance of attentional tasks are closely related to a reduction in the efficiency of executive functions. In particular, the ability to inhibit irrelevant information is reduced: the elderly show poor performance in selective attention, divided attention, and sustained attention tasks [14,15]. In addition, during aging, there are negative alterations in the affective content of the emotions experienced. However, these are probably influenced not only by internal events, such as the perception of reduced physical and cognitive abilities, but also by the inevitable social changes and life events that people must face in old age [16,17,18]. On the other hand, motivation is characterized by the sensation of a need (biological, psychological, or social) and the consequent tendency to reduce tension by satisfying that need. In old age, the balance between desire and consumption activity may be disrupted by a reduction in the availability of behavioral options. In this sense, the elderly often risk attributing their failures to old age, which also affects their performance in cognitive tasks, especially memory: if they believe they cannot effectively deal with a mnemonic task, they will not be motivated to commit themselves, and their performance will be negatively affected [19]. Learned helplessness can thus lead to a vicious circle in which the belief in one’s own inadequacy leads to a decrease in perceived self-efficacy, to avoidance, and to a consequent real deficiency dependent on the lack of mental training [20,21]. Self-efficacy refers to the ability to take a proactive role in one’s own life [22], promoting both the modification of dysfunctional behaviors and the stabilization of virtuous habits [23]. According to Scult and colleagues [24], during aging, self-efficacy can be related to global psychophysical well-being, highlighted by better sleep quality and decreased pain perception. In light of this, promoting the development or strengthening of self-efficacy in the elderly, also through the regular practice of physical activity, could represent an optimal strategy capable of ensuring active aging. The World Health Organization (WHO) has proposed the concept of “active aging” in order to emphasize the optimization of opportunities for the elderly in terms of both their personal and social well-being. From this perspective, the term “health” is not simply the absence of disease, but “a state of complete physical, mental, and social well-being”. The most recent evidence shows how the beneficial effects resulting from sports practice in old age are not only physical but also psychological and cognitive [25,26,27]. Regular physical activity, along with mental training and social integration, is one of the central protective factors for successful aging. Consistent physical exercise makes it possible to maintain or restore high general physical capacity [28]. A study conducted on elderly and frail subjects shows that, after eight weeks of continuous exercise, there is an increase in resistance and muscle strength of up to 180% [29], an increase in flexibility, balance, and coordination [30] with a consequent reduction in the risk of falls [31], and an improvement in speed and in the accuracy of movements [32,33]. Several studies [34,35] highlighted how aerobic fitness practice can improve cognitive function throughout life, highlighting the protective role of regular physical exercise in the cognitive efficiency of elderly subjects. Moreover, numerous studies [36,37] evaluated, in old healthy subjects, the effects of physical activity in terms of motivation, life goals, positive emotions, and the risk of dementia. It was found that those who practiced physical activity at least three times a week had a 32% reduced risk of developing neurodegenerative disorders. Moreover, the results indicated that, after eight years of sporting practice, the subjects showed lower rates of physical disability, fewer physical pathologies, a reduced number of hospitalizations, an increased perceived well-being in terms of improved mood, an increase in the number of life goals, and increased self-efficacy. Given the influence exerted by mood on psychological, cognitive, and social systems, it follows that its improvement due to physical activity can direct the individual towards a virtuous circle that will lead to positive effects in all other areas [38,39,40]. Living in a network of social relationships is also a consistent predictor of good aging, and sports practice favors an expansion of that network [41]. Research has shown that psychosocial factors are closely associated with a deterioration in the overall health of the elderly; in fact, the perception of having lost control and independence over their lives is associated with an important functional deterioration and a sedentary lifestyle, suggesting that the relationship between physical activity and health may have a two-way influence [42]. The simplest and most natural physical activity is walking. Unfortunately, there are not many studies available on the effects of the continued practice of this activity as it relates to aging, while medical research abounds on the benefits of hiking on the physical functionality of many systems [43]. In general, hiking with regularity favors an improvement in general physical fitness [44,45]. In addition, hiking contributes towards the previously described benefits to cognitive abilities, and it is reasonable to assume that the individual who practices sport in a natural setting can benefit from further advantages [46]. Atchley and Strayer (2012) [47] hypothesized that the natural environment itself can have a positive impact on cognitive functions such as selective attention, problem-solving, and inhibition. Their research showed that exposure to natural settings appears to replenish some lower-level modules of the executive attentional system. The authors hypothesized that natural environments are associated with exposure to stimuli that elicit a kind of gentle, soft fascination, and are both emotionally positive and low-arousing. Indeed, hiking seems to be beneficial in terms of psychological well-being, showing therapeutic effects against stress, anxiety, and depression. The increase in psychological well-being due to this sport is also linked to reaching this goal and the consequent increase in self-esteem and perceived self-efficacy [48,49]. Evidence on the benefits of physical activity in general, and particularly on aging, has raised questions about the possibility that there could be differences in terms of cognitive performance and perceived psychological well-being among elderly people practicing intense, moderate, or reduced physical activity [50,51]. Despite the large body of literature supporting the protective role of physical activity in aging populations [52,53,54], it is important to highlight how different sports can modulate cognitive functioning, executive process, and mood. In the present study, we aimed to evaluate whether, and how, three groups with different levels of physical activity (outdoor hiking, indoor gentle exercise and sedentary) differ with respect to both their cognitive performance and psychological profiles. Based on the benefits for global health described above concerning the practice of physical activity in a natural outdoor setting, we expected a better global functioning level in the hiking group participants than those in the other two groups. In fact, performing physical activity in a natural environment can have beneficial effects on cognition, mood, and health, promoting, also during aging, a high level of global daily functioning [55]. According to the biophilia hypothesis [56], during the entire lifespan, humans experience the need to connect and affiliate with the natural environment. When satisfied, a virtuous mechanism with positive psychophysical effects and increased global well-being is triggered [57,58]. Moreover, a large body of studies highlights the positive impact of a natural environment on emotional well-being and global cognitive functioning [59,60]. Typically, hiking is performed in a more natural and immersive setting than gentle gymnastics. Hence, subjects who practice hiking can obtain better cognitive performance and may demonstrate a higher level of global function than subjects who perform an activity that does not require a natural setting to be executed (i.e., indoor physical activity) [61].

## 2. Materials and Methods

### 2.1. Participants

A total of 151 individuals between the ages of 65 and 87 (M = 70.90 years; SD = 0.33) were recruited and then divided into three groups: Hikers (HIK): 50 subjects who practice intense–vigorous physical activity performing outdoor hiking, (M = 71.14 years; SD = 5.31; 30 male); Gentle gymnastics (GYM): 51 subjects who practice moderate physical activity, performing indoor recreational swimming (M = 70.53 years; SD = 5.18; 17 males); Sedentary (SED): 50 subjects who did not perform any physical activity (M = 71.06 years; SD = 5.91; 27 males). The criterion for distinguishing between the different levels of physical activity intensity refers to the types of activity practiced. According to Physical Active Guidelines (2018) [62], jogging, running, swimming laps, or hiking can be considered intense–vigorous activities. Conversely, water aerobics, recreational swimming, or active forms of yoga can be included in moderate-intensity activities. Finally, reduced physical activity can refer to subjects not getting any intense nor moderate physical activity beyond movement from daily life activities.

The recruitment of physically active seniors took place through a collaboration with the University of Third Age and the Club Alpino Italiano (CAI): a formal request was sent to the directors of both associations, who then proceeded to provide contacts of the members who proved willing to participate.

The recruitment of physically active seniors in the first two samples required regular practice of the activity in question (at least 2/3 times a week) as a necessary condition for inclusion in the study. A frequency of 2/3 times a week was chosen as the criterion for the definition of regular physical activity. In the recruitment phase, the participants declared the type of physical activity performed (or sedentary) and the frequency with which they practiced physical activity in the last year. All participants were free of medication at the time of the assessment. We excluded subjects with any known neurological or medical condition that might influence cognition, a history of a developmental disorder, a history of substance or alcohol dependence, or current abuse.

The investigation was approved by the local Institutional Review Board of the University of L’Aquila (#22/2017) and was conducted according to the principles established in the Declaration of Helsinki. Moreover, all participants signed informed consent.

### 2.2. Measures

Psychological questionnaires and cognitive tasks were administered to evaluate some individual differences, such as cognitive reserve, perceived self-efficacy, executive functioning, mood, and daily life functioning. The rationale to use these measures was based on the need to have a sufficiently wide range of psychological and cognitive assessment, and to do this in a non-invasive way, through a brief, psychological battery in order to maximize participation.

Mini-Mental State Examination (MMSE—Italian version; The internal consistency reliability of MMSE ranged between 0.82 and 0.91) [63]. The MMSE is a 30-item tool that allows for a brief screening of the mental state of the individual to be conducted; in this case, it was used to identify and possibly exclude individuals with suspected cognitive impairment. The items investigate temporal and spatial orientation (10 items); memory (immediate and deferred/delayed recall; 6 items); attention and calculation (5 items); objects denomination (2 items); repetition (1 item); execution of a task on written (1 item) and oral (3 items) command; writing a meaningful sentence (1 item); and constructional praxis (1 item). Total scores, obtained from the sum of the correct answers provided by the subject, range from 0–30 and the final score is corrected for age and education. A total score of 25 is considered as normal. Total scores ≤ 24 indicate possible cognitive impairment.

Cognitive Reserve Index Questionnaire (CRIq—Italian version, Cronbach’s alphas = 0.62) [6]. The CRIq is an instrument for the standardized measurement of the cognitive reserve accumulated by individuals through their lifespan. The CRIq includes some demographic data, and 20 items grouped into three sections: education, working activity, and leisure time:CRI-Education: years of education and possible training courses (at least six months). The raw score of this section is the sum of these two values;CRI-Working Activity: five different levels of working activities are available, dealing with the degree of intellectual involvement and personal responsibility (unskilled, manual work, skilled manual work, skilled non-manual or technical work, professional occupation, highly intellectual occupation). The working activity is recorded as the number of years in each profession over the lifespan. The raw score of this section is the result of years of working activity multiplied by the cognitive level of the job (from 1–5);CRI-Leisure Time: cognitively stimulating occupations carried out during leisure time. Sixteen items were related to various intellectual, social, and physical activities. The frequency (i.e., never/rare, often/always) and the number of years (how long each activity had been carried out) were recorded. The raw score of this section is the total number of years of activity for which frequency is often/always.

The average of these three indices (CRI-Education, -Working Activity, and -Leisure Time) constitutes the Cognitive Reserve Index (CRI) score. The final score of the questionnaire and its 3 subscales are standardized and expressed on a scale with M = 100 and SD = 15. CRI can be classified into five levels, Low (less than 70), Medium–Low (70–84), Medium (85–114), Medium–High (115–130) and High (more than 130), respectively. The CRIq instructions and the Excel file for the automatic calculation of subscores are available at https://dpg.unipd.it/en/criq (accessed on 1 July 2020).

General Self-Efficacy Scale (GSE—Italian Version, Cronbach’s alphas between 0.76 and 0.90) [64,65]. The GSE is a tool to measure perceived self-efficacy, and the importance of measuring this construct is evident considering its influence on multiple processes (cognition, motivation, affection, and decision-making). It is designed to assess self-beliefs relating to the ability to cope with a variety of stressful problems in life. It is a psychometric scale composed of 10 items with responses on a 5-point Likert scale, from “I strongly disagree” to “I strongly agree”. The final score is obtained from the sum of the answers provided by the subject (10–50 range), with a higher score indicating more self-efficacy. The administration takes 5–10 min.

Letter Cancellation Test (LCT) [66]. The LCT is a tool to evaluate sustained and selective attention. The subject is required to cross out the target letters (S-H-O) as quickly and accurately as possible in a 5 min time period. The targets are placed in a matrix (36 × 50) containing distracting stimuli (other capital letters) and for each matrix, 300 hits were possible. During the evaluation, the following parameters are calculated and considered dependent variables: the number of completed rows (as index of speed), the number of correct answers, the number of errors, and the number of omissions.

Everyday Competence Questionnaire (ECQ, Cronbach’s alpha = 0.843) [67]. The ECQ is a tool to evaluate the skills and abilities of a subject in carrying out simple activities in daily life. It consists of 17 items: the examiner asks questions and assigns a score based on the answer provided by the subject, according to the item (0–2; 0–3; 0–4). The administration takes 5–10 min, and the scoring allows for 8 different scores to be obtained: leisure activities (LSA), sport (S), subjective well-being (SWB), linguistic abilities (LA), housekeeping (HK), daily routine (DR), manual skills (MS), and mobility (M). The higher the score on each subscale, the better the overall daily functioning. Geriatric Depression Scale, short form (GDS, Cronbach’s alpha = 0.83) [68]. The GDS is a 15-item tool with a dichotomous response (yes/no), and for each item, the relative score is reported based on the answer provided (0–1; range 1–15). A 0 to 5 score indicates no depression; a 6 to 10 score indicates mild depression; a ≥11 score indicates severe depression. Administration of the test takes 5 min.

### 2.3. Procedure

The administration of the questionnaires took place individually and in a single meeting, lasting 30–60 min per subject. An appointment was agreed upon with each of the participants in a secluded and quiet place, during daytime hour, in order to avoid the impairment of cognitive performance due to tiredness or sleepiness. Before the questionnaires were administered, the study’s objectives were briefly described to each subject, and, proceeding in a completely anonymous way, the privacy of the collected data was guaranteed. The administration of each questionnaire was preceded by an explanation of its characteristics and compilation, starting with the cognitive tests and ending with the psychological ones.

### 2.4. Statistical Analysis

For each dependent variable of the MMSE (total score), CRIq (CRI-Education, CRI-Working Activity, CRI-Leisure Time, CRI-Total), the GSE (total score), LCT (number of: completed lines, correct answers, errors, and omissions), ECQ (LSA, S, SWB, LA, HK, DR, MS, M), and GDS (total score), one-way analysis of variance (ANOVA) were run to compare the three groups (HIK, GYM, SED). The same analysis was run for age and education.

The alpha level was fixed to ≤0.05. In the case of significant differences, Tukey’s HSD post hoc test was performed. All statistical analyses were performed using IBM SPSS Statistics for Macintosh, version 25.0 (IBM Corp., Armonk, NY, USA).

## 3. Results

For a clearer and more schematic view of the scores obtained, the reader can refer to Table 1, which shows the mean ± standard deviation of the scores in each questionnaire.

Age and education

For “age” and “education” the one-way ANOVA did not show any significant differences, indicating that the three groups are demographically homogeneous; therefore, they were cognitively and psychologically comparable.

Mini-Mental State Examination

For “MMSE (total score)” the one-way ANOVA showed significant differences between the groups (F2,148 = 18.50; *p* < 0.0000001): Tukey’s HSD post hoc test showed that the HIK group reported a significantly higher score (27.89 ± 1.0; *p* = 0.0003) than the other groups (GYM = 26.32 ± 1.76; SED = 26.20 ± 1.74). However, it should be clarified that none of the groups reported pathological scores and that no subject denoted “clinical” signs of cognitive decay.

Cognitive Reserve Index Questionnaire

CRI-Education. The one-way ANOVA did not show significant differences between the groups, indicating that there were no significant differences in school education, as was also demonstrated by the absence of significant differences in the years of schooling.

CRI-Working Activity. The one-way ANOVA showed significant differences in the Working Activity dimension (F2,148 = 4; *p* = 0.02). Tukey’s HSD post hoc test showed that the HIK (106.54 ± 14.31) and GYM (107 ± 18.04) groups reported significantly lower scores (*p* = 0.01) than the SED group (115.84 ± 22.43), indicating that, considering cognitive reserve, the former groups had less protective work experiences (Figure 1).

CRI-Leisure Time. The one-way ANOVA showed significant differences between groups for the Leisure Time dimension (F2,148 = 16.5; *p* = 0.0000003). Tukey’s HSD post hoc test showed that the HIK group (131 ± 18.83) reported a significantly higher score than both the GYM group (118.1 ± 18.7; *p* = 0.0005) and the SED group (109.28 ± 19.5; *p* = 0.0001), indicating that the leisure activities practiced by the hikers are healthier and potentially protective as a function of aging (Figure 2).

CRI-total. The one-way ANOVA showed significant differences between the groups in the total score of cognitive reserve (F2,148 = 3.4; *p* = 0.036). Tukey’s HSD post hoc test showed that the HIK groups reported a significantly higher score (124 ± 13.43; *p* = 0.05) than the other groups (GYM = 116.63 ± 16.48; SED = 116.82 ± 18.19) (Figure 3).

General Self-Efficacy Scale

The one-way ANOVA did not show significant differences between groups with respect to self-efficacy, indicating that the three groups reported no differences in the perception of their ability to cope with stressful life events.

Letter Cancellation Test

Number of rows. The one-way ANOVA for this subscale showed significant differences between the groups (F2,148 = 20.65; *p* = 0.00000001) and the Tukey’s HSD post hoc test showed that the HIK group was significantly faster (*p* = 0.0003; 21.3 ± 5.7) than both the GYM (13.9 ± 6.3) and the SED (14.3 ± 7.3) groups (Figure 4).

Number of correct answers. This dimension also showed significant differences between the groups (F2,147 = 31.2; *p* < 0.00000001): Tukey’s HSD post hoc test showed that the HIK group was more efficient (129.1 ± 11.4; *p* = 0.0005) than both the GYM (78.9 ± 8.9) and the SED (79.6 ± 11.4) groups.

Number of errors. The one-way ANOVA did not show significant effects for “group” in the “errors” dimension.

Number of omissions. Lastly, the one-way ANOVA showed significant differences between the groups (F2,146 = 3.5; *p* = 0.03). Tukey’s HSD post hoc test showed that the HIK group was significantly less accurate (42 ± 5; *p* = 0.05) than both the GYM (28.6 ± 4.5) and the SED (33.5 ± 5.4) groups.

Everyday Competence Questionnaire

Leisure Activities (LSA). The ANOVA showed significant differences between the groups (F2,148 = 18.72; *p* < 0.0000001). Tukey’s HSD post hoc test showed that the HIK group (11 ± 1.7) reported a significantly higher score than both the GYM (9.3 ± 2.4; *p* = 0.0003) and the SED (8.2 ± 2.6; *p* = 0.005) groups.

Sport (S). The ANOVA showed a significant difference between the groups (F2,148 =183.56; *p* < 0.0000001). Tukey’s HSD post hoc test showed that the SED group (1.02 ± 0.91) reported significantly lower scores than both the GYM (3 ± 0.1) and the HIK (2.9 ± 0.2) groups (*p* = 0.000005).

Linguistic Abilities (LA). The ANOVA showed significant differences between the groups (F2,148 = 4.4; *p* = 0.01): Tukey’s HSD post hoc test showed that the HIK group (3 ± 0.1) reported a significantly higher score (*p* = 0.05) than both the GYM (2.8 ± 0.4) and the SED (2.8 ± 0.4) groups.

Daily Routine (DR). The ANOVA showed significant differences between the groups (F2,148 = 5.36; *p* = 0.005): Tukey’s HSD post hoc test showed that the GYM group (11.9 ± 1.8) scored significantly higher than both the HIK (11.4 ± 1.7; *p* = 0.05) and the SED (10.4 ± 3; *p* = 0.01) groups.

Manual Skills (MS). The ANOVA showed significant differences between the groups (F2,148 = 7.2; *p* = 0.001). Tukey’s HSD post hoc test showed that the HIK group (4.2 ± 1.3) scored significantly higher than both the GYM (3.6 ± 1.3; *p* = 0.03) and the SED (3.1 ± 1.5; *p* = 0.01) groups. Moreover, the GYM group scored higher than the SED group (*p* = 0.05).

Mobility (M). The ANOVA showed significant differences between the groups (F2,148 = 16.7; *p* < 0.000001): Tukey’s HSD post hoc test showed that the HIK group (4.3 ± 1) scored significantly higher than both the GYM (3.3 ± 1.2; *p* = 0.0005) and the SED (2.9 ± 1.4; *p* = 0.001) groups. Moreover, the GYM group scored significantly higher than the SED group (*p* = 0.03).

No other significant effects were found for the Subjective Well-Being and Housekeeping subscales, respectively.

Geriatric Depression Scale

The one-way ANOVA showed significant differences between the groups in the “depression” subscale (F2,148 = 5.74; *p* = 0.004). Tukey’s HSD post hoc test showed that the SED group (3.32 ± 24) reported a significantly higher score than both the GYM (2.63 ± 2.3; *p* = 0.03) and the HIK (1.84 ± 1.8; *p* = 0.01) groups (Figure 5), indicating a higher level of depression. Nevertheless, it is necessary to point out that none of the three groups reported pathological scores.

## 4. Discussion

The groups’ homogeneity concerning the demographic variables allowed us to compare the psychological and cognitive scores measured in this study. The comparison of the scores reported in the MMSE showed a difference in global cognitive function between the groups, with significantly higher scores reported by the HIK group, although none of the three groups showed a pathological decline. Considering the CRIq measures, no differences were found in the Education dimension, while the two groups of active elderly subjects reported significantly lower scores than the SED group in the Work Activity dimension, indicating that the active elderly subjects have had less protective work experiences in relation to cognitive reserve. It is therefore possible to hypothesize that work activity does not affect the quality of cognitive performance, since the HIK group achieved better results in both the attentional and MMSE tasks. As for the Leisure Time dimension, the HIK group reported more healthy and potentially protective activities, suggesting the important role of these activities in preserving cognitive functions during aging. In a previous study [69], a similar effect was found in patients with early Huntington’s disease, in which a higher Leisure Time score was positively associated with better cognitive performance, confirming the protective role of this factor on cognitive functioning, both during physiological aging and in neurological diseases [70,71,72]. The CRIq-total score is also significantly higher in the HIK group than the other two groups. In the present study, no relevant effects on self-efficacy were found. Therefore, it can be concluded that the perception of coping ability is overlapping in the various participants: these data contrast with those found in the literature [73,74].

Instead, interesting data were found in the analysis of the LCT scores. Significant differences were found in: (1) speed execution, since the HIK group completed a greater number of lines; (2) efficiency, as the HIK group provided a greater number of correct answers; and (3) accuracy, considering that the HIK group omitted a greater number of targets than the other groups. Therefore, the greater speed and efficiency shown by the HIK group are accompanied by a greater number of omissions, perhaps due to a motivational orientation towards the goal or due to the greater competitiveness of the subjects who practice this sport, which would make them carry out the test faster, but less accurately. The accuracy–speed ratio describes a phenomenon well-known in the literature: the subjects performing cognitive tasks are instructed to respond as quickly as possible without sacrificing accuracy [75,76,77,78]. On the other hand, many intervening variables (i.e., cognitive impairment, personality traits, sleep quality) can modulate this goal shifting, time by time, in balance towards speed, or accuracy [79,80,81].

Thus, the results showed that the HIK group significantly differed from the other groups in most of the cognitive measures, while the scores obtained by the GYM group did not significantly differ from those obtained by the SED group, where similarities in most of the cognitive tasks were observed. Given the significant results reported by the HIK group, it is possible to assume a better cognitive profile for these subjects. However, it is not possible to define a direction for the relationship between hiking and cognitive functioning and to specify whether hiking improves cognitive skills, or if subjects who show better cognitive abilities have a greater tendency to practice this activity. The direction of this relationship may be clarified by future studies.

Considering the ECQ results, the HIK group reported significantly higher scores in most of the subscales, above all showing greater autonomy in leisure activities and greater engagement in social activities than the other groups. This last aspect may be related to the fact that hiking, in our experimental group, is often practiced in groups. These subjects have often reported participating in social activities through conferences, dinners, and events. Consistent with recruitment requirements, the GYM and the HIK groups reported regular training, while the SED group obtained a significantly lower score in the sport subscale. The HIK group members were also more eloquent than the others, while the GYM group members were more flexible in organizing their days and more autonomous in everyday life, whereas both the HIK and SED group members were more dependent and routinary. In order to interpret these data, it must be considered that the items of the daily routine scale refer largely to domestic activities (i.e., cooking, cleaning), and most of the HIK group subjects were men, while the GYM group presented the opposite pattern; in fact, in the age range considered in this study, male subjects often reported relying on their partner to carry out housework. The HIK group also reported higher scores in both the mobility and manual skills subscales. The latter factor can relate to the low score reported by this group in the working activities subscale of the CRIq: many of these subjects reported work activities characterized by a low degree of cognitive commitment and greater physical and manual commitment (i.e., laborers, farmers, gardeners, plumbers). Lastly, the subjective well-being scale scores were not significant, since most of the subjects of the entire sample provided average answers. The protective role of physical activity on mood during aging is extensively described in the literature and in line with our results [82,83]: sedentary people reported higher scores, followed by the GYM group and lastly the HIK ones. Hence, the protective role played by physical activity, especially in a natural environment, on the onset of depressive conditions in aging seems to be confirmed. However, it should be specified that none of the three groups reported pathological scores. Taken into consideration the points discussed above, it is appropriate to make a remark: the data from our study seem to show how the regular practice of intense-vigorous physical activity performed in a natural environment, such as hiking, is associated with better cognitive functioning when compared to sedentary behaviors and in the group who practice indoor physical activity characterized by a moderate intensity. However, in addition to the type and intensity of physical activity, the better cognitive performance observed in the HIK group could depend on other factors that are able to modulate this effect. Among other factors, personality and temperament features, physical and social functioning, environmental variables, as well as the quality of social relationships can certainly play a mediating role.

Nonetheless, the present study presents some limitations that should be kept in mind when we try to interpret the observed data. First, to better explore the complex relationship between physical activity practice, cognitive processes, and executive functioning, future studies will need to implement a more extensive cognitive assessment battery that could include measures of mental set-shifting processes, working memory, behavioral and cognitive flexibility, as well as that of attentional switching. Moreover, a limit could be represented by the reduced sample size, although this was sufficiently reliable. In light of this, we may reserve the right to increase the number of participants in a future project. Another limit could be the different time of day in which the administrations took place. Although the authors tried to avoid running the tests in the evening in order to prevent negative influences on performance caused by tiredness or sleepiness, the GYM subjects often requested to fix the meeting immediately after their training at the gym, which may have negatively affected the cognitive results. Another potential limitation could be related to the motivational factors associated with an individual tendency to practice one activity rather than another. For example, the subjects who practice gentle gymnastics could have been pushed to physical activity for social or medical reasons, rather than an autonomous and spontaneous choice based on their nature, interests, and desires. On the contrary, the choice to practice a more demanding activity, such as hiking, could be the result of a greater competitive predisposition, it could depend on a family tradition, or on a passion for being outdoors, in a natural environment. Finally, the way in which the participants were recruited should be carefully considered. The HIK group consisted mainly of members of the CAI, and GYM group members were recruited from the University of the Third Age. Consequently, these groups may possess specific characteristics that do not make them representative of larger populations of hikers and those participating in elderly gymnastics. A possible future investigation could aim to reduce the influence of these intervening variables through a randomized sampling within these populations.

## 5. Conclusions

In conclusion, the HIK group appears to differ from the other two groups for most of the measurements carried out, presenting better cognitive performance and a specific psychological profile. On the contrary, the GYM group seems to have a greater affinity with the SED group than with the HIK one, both cognitively and psychologically. It may be that this difference is due to the different environments in which the physical activity is practiced: the HIK group walks through nature, which, considering the literature, seems to be a protective factor against cognitive and psychological decline. Instead, the GYM group is practicing an indoor activity that promotes good aging, but not as much as an outdoor activity. Future developments of this study could specifically investigate why the GYM groups did not show the expected benefits based on what previous studies have reported.

Finally, further investigations confirming the benefits of physical activity in a natural environment will be able to offer a positive cue for design projects aimed at promoting well-being and healthy lives within the communities.

## Figures and Tables

**Figure 1 brainsci-13-01581-f001:**
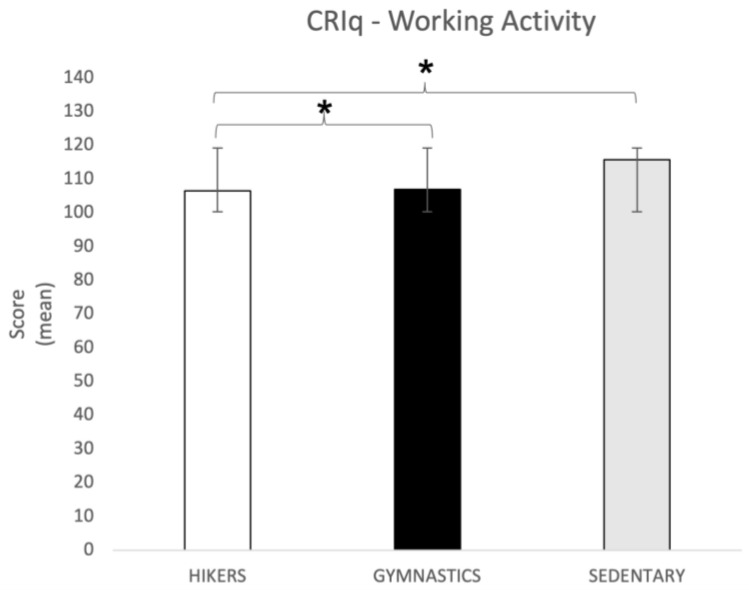
CRIq working activity score (mean ± SD) in all three groups. Post hoc comparison: * *p* = 0.001. CRIq = Cognitive Reserve Index Questionnaire.

**Figure 2 brainsci-13-01581-f002:**
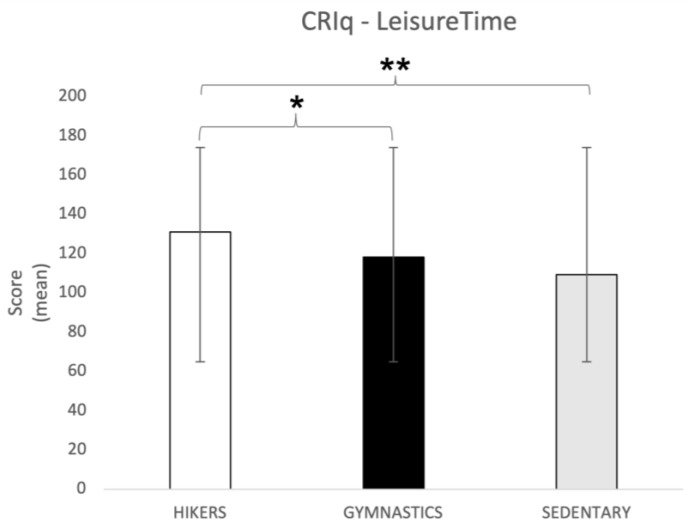
CRIq leisure time score (mean ± SD) in all three groups. Post hoc comparison: * *p* = 0.0005; ** *p* = 0.0001. CRIq = Cognitive Reserve Index Questionnaire.

**Figure 3 brainsci-13-01581-f003:**
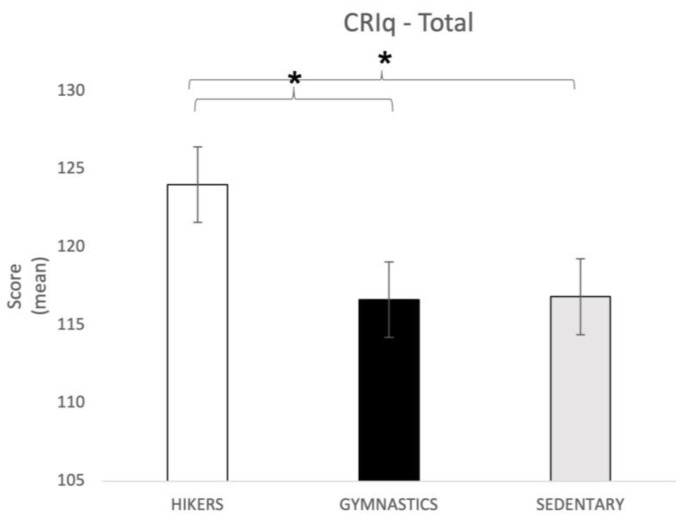
CRIq total score (mean ± SD) in all three groups. Post hoc comparison: * *p* = 0.05. CRIq = Cognitive Reserve Index Questionnaire.

**Figure 4 brainsci-13-01581-f004:**
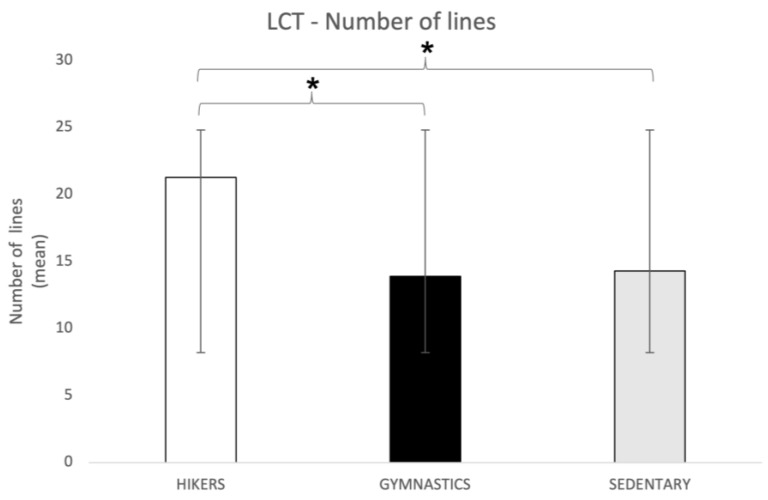
LCT number of complete lines (mean ± SD) in all three groups. Post hoc comparison: * *p* = 0.003. LCT = Letter Cancellation Test.

**Figure 5 brainsci-13-01581-f005:**
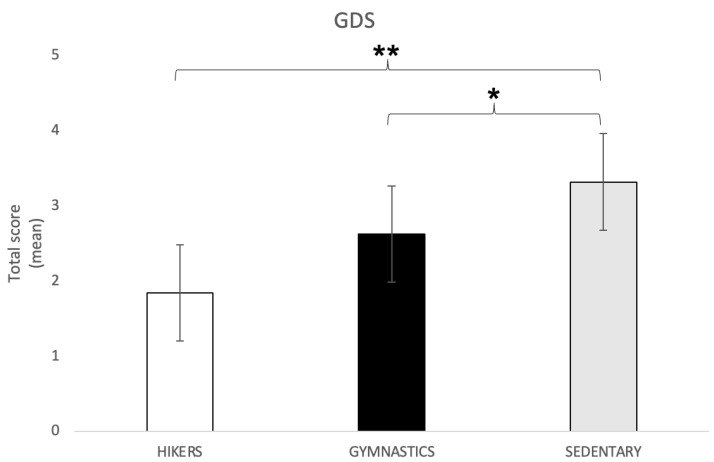
GDS total score (mean ± SD) in all three groups. Post hoc comparison: * *p* = 0.03; ** *p* = 0.01; GDS = Geriatric Depression Scale.

**Table 1 brainsci-13-01581-t001:** Demographic and psychological scores (mean ± standard deviation) in all groups.

		Hikers (M ± SD)	Gentle Gymnastics (M ± SD)	Sedentary (M ± SD)
**AGE**	Years	71.14 ± 5.31	70.52 ± 5.18	71.06 ± 5.90
**EDUCATION**	Years	12.64 ± 3.71	13.13 ± 3.69	12.92 ± 4.14
**MMSE**	Total	27.89 ± 0.98	26.32 ± 1.78	26.19 ± 1.73
**CRIq**	Education	116.68 ± 13.21	111.23 ± 16.99	112.54 ± 12.60
	Working Activity	106.54 ± 14.30	107.00 ± 18.04	115.84 ± 22.42
	Leisure Time	131.00 ± 18.83	118.09 ± 18.73	109.28 ± 19.48
	Total	124.00 ± 13.43	116.62 ± 16.48	116.82 ± 18.19
**GSE**	Total	39.98 ± 6.23	41.15 ± 5.71	38.34 ± 6.30
**LCT**	Completed Lines	21.28 ± 5.73	13.90 ± 6.26	14.30 ± 7.31
	Correct Answers	129.08 ± 11.36	78.89 ± 8.88	79.60 ± 8.92
	Errors	0.98 ± 1.31	1.24 ± 1.29	1.38 ± 1.79
	Omissions	42.04 ± 5.03	28.61 ± 4.52	33.5 ± 5.43
**ECQ**	Leisure Activities	10.96 ± 1.68	9.29 ± 2.44	8.18 ± 2.61
	Sport	2.94 ± 0.23	2.98 ± 0.14	1.02 ± 0.97
	Subjective Well-Being	3.06 ± 0.68	2.84 ± 0.67	2.74 ± 0.87
	Linguistic Abilities	3.00 ± 0.11	2.84 ± 0.36	2.82 ± 0.43
	House Keeping	6.63 ± 1.32	6.72 ± 1.45	5.96 ± 2.18
	Daily Routine	11.38 ± 1.68	11.88 ± 1.81	10.42 ± 3.07
	Manual Skills	4.18 ± 1.35	3.64 ± 1.27	3.14 ± 1.48
	Mobility	4.26 ± 0.96	3.33 ± 1.17	2.92 ± 1.38
**GDS**	Total	1.84 ± 1.84	2.62 ± 2.26	3.32 ± 2.41

MSSE = Mini Mental State Examination; CRIq = Cognitive Reserve Index Questionnaire; GSE = General Self-Efficacy Scale; LCT = Letter Cancellation Test; ECQ = Everyday Competence Questionnaire; GDS = Geriatric Depression Scale.

## Data Availability

The data that support the findings of this study are available from the corresponding author upon request.

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
