# Peer review of "Effects of Physical Activity on Cognitive Functioning: The Role of Cognitive Reserve and Active Aging"

_brainsci, 2023, doi:10.3390/brainsci13111581_

Round 1

Reviewer 1 Report

Comments and Suggestions for Authors

This study investigated the modulation of cognitive skills by the intensity of physical training. The number of subjects appears adequate, the methods are o.k, and the results are clear.

l.17: the second comma must be replaced by a point.

l.148 f Participants. The criteria for intense, moderate and reduced PA should be specified.

l.170 f measures: the applied methods were not mainly questionnaires, but also cognitive tests such as MMSE and LCT. This should be clearly stated.

Unfortunately besides the MMSE which is not really appropriate for this group (as all subjects yielded high levels), the authors did only administer one neuropsychological test, namely the LCT. Here the Hikers were much faster but less accurate than the other two groups

Accordingly the authors stated that the results could not prove a better cognitive profile of the hikers than the other groups.  The lack of specific cognitive tests (e.g. for attention memory, executive functions), is, to my opinion, the strongest limitation of the study. This should be mentioned in the discussion.

Comments on the Quality of English Language

ok; minor chances possible

Author Response

This study investigated the modulation of cognitive skills by the intensity of physical training. The number of subjects appears adequate, the methods are o.k, and the results are clear.

Authors’ Answer (AA): We thank the Reviewer for his/her encouraging opinion on our manuscript.

Point 1: l.17: the second comma must be replaced by a point.

AA: Thank you: in the revised version of the manuscript, we made the suggested replacement.

Point 2: l.148 f Participants. The criteria for intense, moderate and reduced PA should be specified.

AA: We thank the Reviewer for his/her comment. According to the present suggestion (and another one raised by Reviewer #2), in the revised version of the manuscript we tried to clarify this aspect and related issues.

Poin 3: l.170 f measures: the applied methods were not mainly questionnaires, but also cognitive tests such as MMSE and LCT. This should be clearly stated.

AA. The Reviewer is right. In the revised version of the manuscript, this sentence has been changed and a specification has been included in the text.

Point 4: Unfortunately besides the MMSE which is not really appropriate for this group (as all subjects yielded high levels), the authors did only administer one neuropsychological test, namely the LCT. Here the Hikers were much faster but less accurate than the other two groups.
Accordingly the authors stated that the results could not prove a better cognitive profile of the hikers than the other groups. The lack of specific cognitive tests (e.g. for attention memory, executive functions), is, to my opinion, the strongest limitation of the study. This should be mentioned in the discussion.

AA: We thank the reviewer for the opportunity to clarify this aspect. We agree with him/her regarding the need to use other cognitive tasks that, in addition to the assessment of selective and sustained attention (LCT), could shed light on other executive processes (i.e., working memory, cognitive and behavioral flexibility, planning, and/or mental set-shifting). When the study was carried out our first aim was to obtain as much data as possible, by avoiding boring participants and overloading them with too many tasks/questionnaires. Moreover, an extension of the cognitive battery cannot be done, at this moment. Therefore, in the revised version of the manuscript, we explicitly highlighted this limitation.

Point 5: Comments on the Quality of English Language ok; minor chances possible

AA. We thank the Reviewer for his/her comment. According to this, the current version of the manuscript has been checked by a native English speaker.

Reviewer 2 Report

Comments and Suggestions for Authors

The manuscript entitled “Effects of Physical Activity on Cognitive Functioning: The Role of Cognitive Reserve and Active Ageing” is an interesting and valuable contribution to the literature on aging. In the manuscript, the authors detail the measurement of the cognitive performance of 151 seniors organized into three groups based on the intensity of their habitual physical activity: intense (i.e., hikers), moderate (i.e., gentle gymnastics), and low (i.e., inactive). In my modest opinion, the following are these concerns:

In the introductory section, the authors state “[i]n the present study, we aimed to evaluate whether and how the three hiking, gentle exercise, and sedentary groups differ with respect to both cognitive performance and psychological profile. Specifically, we expected a better global functioning level in hiking than the other two groups”. This hypothesis has to be supported by a clearly stated and robust rationale.

The groups labeled as performing either intense or moderate physical activities may differ not only in the degree of the physical activity enacted but also in its quality. Consider also the context in which “gentle gymnastics” and hiking may occur.  In my modest opinion, the authors need to provide a detailed operational definition of “gentle gymnastics” and “hiking”. How often do these activities are performed? What is “gentle gymnastics”? Where is it performed? Indeed, the setting of one’s physical activity may be relevant if one considers the biophilia hypothesis.

The rationale for each of the individual difference measures selected by the authors is not clearly stated.  

The sample is rather small given the within-group variability of the selected measures. Should bootstrapping be used?

Are the samples adequate for carrying out parametric statistics? If the normality assumption is unlikely to apply, should non-parametric statistics be used to assess group differences?  

In light of the fact that these three groups’ lifestyles may differ above and beyond the intensity of the physical activity they habitually perform, it may be reasonable to consider a more careful interpretation of the data. Cognitive differences may be attributed to differences in other aspects of the participants’ lifestyles.

Comments on the Quality of English Language

Minor editing of the English language may be advised. 

Author Response

The manuscript entitled “Effects of Physical Activity on Cognitive Functioning: The Role of Cognitive Reserve and Active Ageing” is an interesting and valuable contribution to the literature on aging. In the manuscript, the authors detail the measurement of the cognitive performance of 151 seniors organized into three groups based on the intensity of their habitual physical activity: intense (i.e., hikers), moderate (i.e., gentle gymnastics), and low (i.e., inactive). In my modest opinion, the following are these concerns:

Authors’ Answer (AA): We thank the Reviewer for his/her encouraging opinion on the manuscript.

Point 1: In the introductory section, the authors state “[i]n the present study, we aimed to evaluate whether and how the three hiking, gentle exercise, and sedentary groups differ with respect to both cognitive performance and psychological profile. Specifically, we expected a better global functioning level in hiking than the other two groups”. This hypothesis has to be supported by a clearly stated and robust rationale.

AA: We thank the Reviewer for this comment. In the revised version of the manuscript, we described more in depth these aspects, also to cope with the next point raised by the Reviewer.

Point 2: The groups labeled as performing either intense or moderate physical activities may differ not only in the degree of the physical activity enacted but also in its quality. Consider also the context in which “gentle gymnastics” and hiking may occur. In my modest opinion, the authors need to provide a detailed operational definition of “gentle gymnastics” and “hiking”. How often do these activities are performed? What is “gentle gymnastics”? Where is it performed? Indeed, the setting of one’s physical activity may be relevant if one considers the biophilia hypothesis.

AA: We thank the Reviewer for this comment. In the previous version of the manuscript, we implicitly considered the relevance of contextual issues, without properly discuss and introduce them to the reader. According to this, in the revised version of the manuscript (both in the introduction and the methods), we tried to better specify these aspects, also introducing the “Biophilia hypothesis”.

Point 3: The rationale for each of the individual difference measures selected by the authors is not clearly stated.

AA: The Reviewer is right: a brief explanation of the rationale of the administered battery has been included in the materials’ section, before the description of each measure.

Point 4: The sample is rather small given the within-group variability of the selected measures. Should bootstrapping be used?

AA: Basically, when we firstly planned and then performed the study, we calculated the more appropriate sample size (see point #5) and a sample of 50 homogeneous participants for each group appeared to be statistically relevant. This statistical “goodness” supported us in applying a classical parametric analysis. With respect to the suggested technique, bootstrapping is a powerful tool that needs to be used carefully; as observed by Hinkley (1994), bootstrapping is very vulnerable to some serious risk of inconsistency if a “wrong” estimator is used. On the light of both these considerations, we used more “canonical” methods of statistical analyses.

Point 5: Are the samples adequate for carrying out parametric statistics? If the normality assumption is unlikely to apply, should non-parametric statistics be used to assess group differences?

AA: We thank the Reviewer for this relevant question. As briefly introduced in the previous point, before the experiment, we calculated the sample size by applying the following standard formula that is valid for the calculation of sample number when we deal with quantitative variables:

Immagine che contiene Carattere, calligrafia, bianco, linea

Descrizione generata automaticamente

Here, the value of σ2 corresponds to the standard deviation of the effect estimated on the basis of data previously published in the literature, the difference μ1-μ2 is related to the difference expected in the study in question for the effect to be satisfactory and qualitatively significant, while the values a e b concern the level of significance and the power of the study (or type I and II error).
The calculation of the sample size was carried out taking into account the values known in the literature and expected in the present project of the MMSE variable, that in the literature is seen as a gold standard for global cognitive functioning assessment, and in the present study can be intended as a good indicator of a potential effect induced by physical activity on cognitive effectiveness.
As regards the values σ
2 2 e μ1-μ2, a double strategy was followed: an "optimistic" calculation of the sample size (assuming the smallest possible sample capable of highlighting a significant effect) and a "pessimistic" one (assuming the highest sample needed to show an equally significant effect). The values a e b, on the other hand, were set in both cases at 0.05 and 0.90.
In the "optimistic" version, replacing the values of variance and estimates of the differences between groups, the value of n resulted to be equal to 39.4 subjects for each group/condition; in the "pessimistic" calculation instead the n requested was equal to 52.6 subjects for each group/condition. Thus, technically and statistically speaking the group investigated in this study would be considered as “sufficiently” representative of the population.
Nonetheless, since this esteem has been done on the basis of previous literature (that could in turn to be biased) we believe that a greater numerosity of groups could give more stability to the data and guarantee more representativeness to the results. As a consequence, we decided to consider this aspect as a potential limit to the study.

Point 6: In light of the fact that these three groups’ lifestyles may differ above and beyond the intensity of the physical activity they habitually perform, it may be reasonable to consider a more careful interpretation of the data. Cognitive differences may be attributed to differences in other aspects of the participants’ lifestyles.

AA: We thank the Reviewer for this comment. We agree with this reflection and according to this in the current version of the discussion, we further described possible factors that can modulate the observed cognitive differences. Undoubtedly, as described in the manuscript, future studies will need to clarify the direction and the causality of such effects.

Point 7: Comments on the Quality of English Language. Minor editing of the English language may be advised.

AA. We thank the Reviewer for his/her comment. According to this, the current version of the manuscript has been checked by a native English speaker.

Round 2

Reviewer 2 Report

Comments and Suggestions for Authors

In my modest opinion, the authors have adequately addressed the concerns expressed in earlier reviews of the manuscript. Thus, the manuscript merits publication. 

Comments on the Quality of English Language

Minor editing of the English language is required.